# Recurrent Nasopharyngeal Cancer: Critical Review of Local Treatment Options Including Recommendations during the COVID-19 Pandemic

**DOI:** 10.3390/cancers12123510

**Published:** 2020-11-25

**Authors:** Michaela Svajdova, Marian Sicak, Pavol Dubinsky, Marek Slavik, Pavel Slampa, Tomas Kazda

**Affiliations:** 1Department of Radiation and Clinical Oncology, Central Military Hospital—Teaching Hospital Ruzomberok, 034 01 Ruzomberok, Slovakia; 2Department of Radiation Oncology, Faculty of Medicine, Masaryk University, 625 00 Brno, Czech Republic; slavik@mou.cz; 3Department of Otorhinolaryngology and Head and Neck Surgery, Central Military Hospital—Teaching Hospital, 034 01 Ruzomberok, Slovakia; sicakm@uvn.sk; 4Department of Radiation Oncology, East Slovakia Oncology Institute, 041 91 Kosice, Slovakia; dubinsky@vou.sk; 5Faculty of Health, Catholic University Ruzomberok, 034 01 Ruzomberok, Slovakia; 6Department of Radiation Oncology, Masaryk Memorial Cancer Institute, 656 53 Brno, Czech Republic; slampa@mou.cz (P.S.); tomas.kazda@mou.cz (T.K.)

**Keywords:** nasopharyngeal cancer, recurrence, salvage surgery, re-irradiation, toxicity

## Abstract

**Simple Summary:**

Options for the curative treatment of locally recurrent nasopharyngeal carcinoma include surgery or re-irradiation. Both approaches have been scientifically explored, yet there is no consensus on the indication or definitive preference of the above two salvage treatments. The aim of this review is to summarize the current evidence on the local treatment of recurrent nasopharyngeal carcinoma. The feasibility, safety, and efficacy of salvage surgery and radical re-irradiation are discussed. Recommendations on treatment modifications during the coronavirus disease 2019 pandemic are included as well.

**Abstract:**

Recurrent nasopharyngeal carcinoma represents an extremely challenging therapeutic situation. Given the vulnerability of the already pretreated neurological structures surrounding the nasopharynx, any potential salvage retreatment option bears a significant risk of severe complications that result in high treatment-related morbidity, quality of life deterioration, and even mortality. Yet, with careful patient selection, long-term survival may be achieved after local retreatment in a subgroup of patients with local or regional relapse of nasopharyngeal cancer. Early detection of the recurrence represents the key to therapeutic success, and in the case of early stage disease, several curative treatment options can be offered to the patient, albeit with minimal support in prospective clinical data. In this article, an up-to-date review of published evidence on modern surgical and radiation therapy treatment options is summarized, including currently recommended treatment modifications of both therapeutic approaches during the coronavirus disease 2019 pandemic.

## 1. Introduction

Nasopharyngeal carcinoma (NPC) is a rare type of head and neck cancer, arising from the posterolateral nasopharynx or pharyngeal recess (fossa of Rosenmüller). It differs significantly from other cancers of the head and neck in epidemiology, causes, clinical behavior, treatment response, and prognosis. A total of 129,000 incident cases and 73,000 deaths were reported in 2018 worldwide [1]. While rarely observed in the United States or Europe, NPC is endemic in Southeast Asia, North Africa, and the Arctic, with the endemic form being invariably associated with prior Epstein–Barr virus (EBV) infection [2] and with undifferentiated nonkeratinizing squamous cell carcinoma (World Health Organization—WHO type III) being the predominant histology [1,3]. Epidemiological trends during the past decade suggest that mortality from the disease has decreased substantially [2]. Clinical use of EBV deoxyribonucleic acid (DNA) analysis as a surrogate biomarker in nasopharyngeal carcinoma continues to increase, with a quantitative assessment of circulating EBV DNA used for population screening, prognostication, and disease surveillance [4,5].

Although it is unclear if the strong link to a viral etiology explains the intrinsic sensitivity of these tumors to radiation therapy (RT), NPC represents a highly chemo- and radiosensitive disease, and upfront therapy usually consists of the highly conformal intensity-modulated radiation therapy (IMRT) and concurrent cisplatin [6]. Following chemoradiotherapy (CRT) as a primary treatment, excellent 5-year local control (LC) rates up to 85% have been observed [7]. However, approximately 10–20% of patients will develop local or regional recurrence after primary therapy [7,8]. Early detection of the recurrence is crucial as the locoregionally recurrent disease is still potentially curable.

A comprehensive search was made of PubMed, Medline, Scopus, and UpToDate using the descriptors “recurrent nasopharyngeal carcinoma”, “salvage surgery”, and “re-irradiation”. The abstracted literature was reviewed, as were references and related material. Only articles published in English that contained at least 10 patients retreated with curative intent were included. Reports describing palliative surgical or radiation retreatment were excluded. Treatment parameters including the use of salvage surgery and the type of surgical approach, technique, dose, and fractionation of re-irradiation and data on severe (≥grade 3) toxicity were abstracted and summarized. 

Considering surgical treatment, only the studies that described the use of endoscopic and maxillary swing nasopharyngectomies were included in Table 1. Studies that described the use of any of the approaches mentioned above but did not precisely evaluate the treatment response with regard to the specific type of surgical approach and instead evaluated the treatment response for diverse surgical approaches altogether were revised as well, but were not included in Table 1. Data on surgical outcome (clear microscopic margins, local control, overall survival), severe treatment-related complications, and treatment-related deaths are summarized in Table 1. Considering re-irradiation treatment, studies that mentioned the use of IMRT, with regard to the country of origin of this study, and evaluated treatment response (complete response, local control, overall survival) within a follow-up of at least 10 months, were included in Table 2. Studies that described the use of non-IMRT techniques (stereotactic radiosurgery, stereotactic body radiotherapy, proton beam therapy, carbon ion radiation therapy, interstitial brachytherapy with radioactive gold or iodine grain implantation or high-dose-rate intracavitary brachytherapy) that evaluated treatment response (complete response, local control, overall survival) are summarized in Table 3. Data on severe toxicity and treatment-related deaths are compiled for both IMRT and non-IMRT techniques in Table 2 and Table 3. Re-irradiation studies that reported the use of any of the above-mentioned radiation treatment techniques but evaluated the treatment response for different techniques altogether were excluded from Table 2 and Table 3.

## 2. Early Detection of the Recurrence and Restaging

Early detection of recurrent disease is of paramount importance. The initial assessment is better carried out by rigid nasopharyngeal endoscopy; according to prospective trials comparing this procedure to computed tomography (CT), a significantly better agreement with the histological findings has been reported [9,10]. The rigid endoscopic examination provides a better view than flexible endoscopy, to the extent that even submucosal bulges can be seen [11]. Narrow-band imaging (NBI) increases the diagnostic value of endoscopy and differentiates early nasopharyngeal recurrences on the basis of distinctive morphological characteristics of mucosal capillary vessels in such lesions [12]. A recent meta-analysis provided reliable evidence that the level of plasma EBV DNA, as a tumor marker for NPC with high sensitivity and specificity, has a high diagnostic value in the detection of recurrent NPC (rNPC), and monitoring of the levels of plasma EBV DNA is recommended in the follow-up of this disease [13]. Magnetic resonance imaging (MRI), aside from providing better images of soft tissues, also shows tumor size and extension to the base of the skull and intracranial cavity.

Prior to any optimal treatment decision-making, a precise restaging is absolutely necessary, as up to 56% of all patients diagnosed with locally recurrent NPC may suffer from a synchronous distant disease [6,7,14,15]. The cumulative incidence of distant failure exceeds 25% even for recurrent T1 tumors on Tumor-Node-Metastases (TNM) classification [16]. The restaging workup should include a whole-body positron emission tomography/computed tomography (PET/CT) scan to exclude distant metastases. In the presence of negative findings on both PET/CT and MRI, the regional nodal and distant disease can be reasonably, although not absolutely, excluded.

## 3. Patient Selection for Retreatment with Curative Intention

A carefully selected subset of patients with rNPC may achieve long-term survival after local retreatment. However, the location of the nasopharynx and its proximity to vital organs and radioresistance induced by previous RT make the therapy challenging. Significant acute and late toxicity must be expected.

### Factors Affecting Prognosis

The exact definition of prognostic factors may guide the provision of individualized treatment and lead to a higher chance of local salvage. The first ever validated prognostic index for risk stratification in patients with radioresistant NPC who present with an isolated local recurrence following primary RT was developed by two Asian academic institutions on the basis of a training cohort of 251 patients and a validation cohort of 307 patients who had undergone salvage treatment using IMRT from 2001 to 2015 [17]. Clinical indices (factors) of retreatment using IMRT were independently validated for survival and grade 5 toxicity-free rate. Covariates that were adversely associated with overall survival (OS) in the training cohort were, in ascending order of weight: larger gross tumor volume (GTV) of the recurrence (*p* = 0.001), higher age at the time of recurrence (*p* = 0.008), repeated IMRT equivalent dose in 2-Gy fractions (EQD2) of ≥68 Gy (*p* = 0.03), prior RT-induced grade ≥3 toxicity (*p* = 0.001), and T3-T4 classification of the recurrent disease (*p* = 0.005). The prognostic index was able to dichotomize the cohort into low-risk and high-risk subgroups that were correlated with survival outcomes and treatment-related mortality after salvage IMRT [17]. Low-risk patients with an estimated 3-year overall survival (OS) in 69–74% represent ideal candidates for curative re-irradiation using IMRT, whereas novel clinical trials are recommended in the unfavorable high-risk subgroup with the reported 3-year OS in 15–31% only. To encourage its widespread clinical use, the prognostic index presented herein is a publicly available online tool at http://prancis.medlever.com [18].

Factors with a major impact on OS in the recursive partitioning analysis (RPA) published by Ward et al. in 2018 with 10.2% of the patients having rNPC included operability (inoperable vs. operable, HR 1.8, 95% CI 1.3–2.5, *p* = 0.0001), presence of organ dysfunction (yes vs. no, HR 1.6, 95% CI 1.2–2.1, *p* = 0.0003), time from previous RT (<2 years vs. ≥2 years, HR 0.9, *p* = 0.028), and Karnofsky performance status (KPS) score (<70 vs. ≥ 70, HR 0.6, 95% CI 0.4–0.9, *p* = 0.025) [19].

The volume of the recurrent tumor is another independent prognostic factor. Five-year OS rates are poorer in patients with a tumor volume >38 cm^3^ than in patients with a tumor volume ≤38 cm^3^ (30.1% vs. 55.9%, *p* < 0.001) [20]. World Health Organization histologic type also determines the outcome in rNPC. Locoregional progression-free survival (LRPFS, *p* < 0.035) and OS (*p* < 0.0001) are both significantly better for patients with WHO type III disease than with WHO type I or II disease [21].

The prognostic significance of biomarkers, including pretreatment circulating plasma EBV DNA titer [22,23,24] and the neutrophile-to-lymphocyte ratio (NLR) [25,26] that serve as biologic surrogates of occult metastases, is uncertain in locally or regionally recurrent NPC. However, studies have shown that a detectable circulating EBV DNA following upfront CRT is associated with poor prognosis of EBV-associated NPC, and, conversely, patients with undetectable circulating EBV DNA may have an excellent prognosis [5,24]. Furthermore, in a retrospective study of 552 patients from an endemic region, the patients with a low NLR (≤2.29) reached a significantly worse survival in comparison to the patients with a higher NLR (>2.29) (HR, 1.46; 95% CI, 1.14–1.85; *p* = 0.002) [27].

## 4. Surgical Treatment

Salvage surgery can be an option for carefully selected patients with a local recurrence or an isolated relapse in the neck [28,29]. A new surgical staging system of recurrent nasopharyngeal carcinoma has been proposed to help determine the therapeutic outcome in resected patients. The “surgical” stage model is based on the recurrent Tumor-Node-Metastases (rTNM) staging system for rNPC and classifies local recurrences as minimal (stage I), limited (stage II), extensive (stage III), and disseminated (stage IV). The survival curves are distinctly separated within these stages (5-year OS 72.0%, 55.1%, 21.1%, and 10.1% for surgical stages I, II, III, and IV, respectively; *p* < 0.001) [30].

Three-year survival rates up to 60% after salvage surgery have been reported, with survival benefit being most significantly observed in patients with T1 and T2 recurrent disease on TNM classification [31,32,33,34]. Locally advanced stages with skull base, cranial nerve, dural or brain involvement, positive surgical margins, and concurrent nodal metastases are associated with poor prognosis [35,36].

Some evidence suggests that surgical salvage may be superior to re-irradiation using IMRT (re-IMRT) in terms of LC, OS, quality of life (QoL), and even treatment cost [34,37]. In a case-matched cohort analysis, surgical treatment was associated with significantly better OS compared to re-IMRT (5-year OS 77.1% vs. 55.5%, *p* = 0.003), higher QoL (mean global health status score 57.6 vs. 29.8, *p* < 0.001), and decrease in severe post-treatment complications (12.5% vs. 65.3%, *p* < 0.001) [37]. To date, there is no consensus on the indications or definitive preference of either surgery or re-irradiation in this clinical setting [38].

### 4.1. Recommendations for Surgical Treatment during the COVID-19 Pandemic

In general, curative treatment of any cancer is challenging not only for the patient but also for the healthcare system, with physicians and other healthcare providers in the pivotal role. This is highly relevant during any era of limited resources. The current coronavirus disease 2019 (COVID-19) pandemic is a real example of a situation where even developed countries with an advanced healthcare system quickly reach a state of limited resources. For this reason, the discussion and guidance of treatment approaches during limited resources is becoming the new unmet medical need. In this article, we held on the current COVID-19 situation, although some parts of the following discussion may be generalizable.

Severe acute respiratory syndrome coronavirus 2 (SARS-CoV-2) is the causative agent of the COVID-19 pandemic with the first reported outbreak in China in 2019. It is found to a high extent in the upper respiratory tract mucosa, especially in the nasopharynx [39]. A high risk of infection is imposed on all head and neck surgeons when treating patients with rNPC. There is general concern over personal protective equipment shortage rises as the number of confirmed COVID-19 cases grows worldwide [40].

All patients should be in a home isolation or, if feasible, hospital quarantine at least 14 days prior to any surgical salvage within the nasopharynx. Two sets of real-time reverse transcription polymerase chain reaction (rRT-PCR) tests for the qualitative detection of nucleic acid from SARS-CoV-2 should be collected 24 hours apart. If the individual remains asymptomatic during the 14 days of isolation or quarantine followed by two separate negative rRT-PCR results, surgery can be performed with standard precautions [41,42,43]. The hospital infection control team, in cooperation with microbiologists and pulmonary physicians, should consult further management of the patients who are tested positive for COVID-19. It is recommended that surgery be postponed until symptoms have resolved and two separate rRT-PCR tests have remained negative. According to some authors, a 12-week waiting time is acceptable for patients with rNPC during the COVID-19 pandemic [40].

### 4.2. Endoscopic Nasopharyngectomy

A worldwide shift from an open to a minimally invasive endoscopic approach is widely observed in the surgical treatment of rNPC [38]. Indications for endoscopic nasopharyngectomy (ENP) using a transnasal approach include early-stage local recurrences (T1, T2 on TNM classification) located at the nasopharyngeal roof that invade minimally to the parapharyngeal space. The major goal of surgery is to achieve microscopically negative margins while preserving the surrounding fragile neurological and vascular structures. With a careful patient selection, ENP may achieve results comparable to re-IMRT, with a reduced rate of severe treatment-related complications [37,44,45,46,47,48,49,50,51,52,53].

Using the modern three-dimensional (3-D) endoscopic navigation system, resection of locally advanced tumors located at the base of the skull is facilitated, with an easier approach to the adjacent pterygopalatine fossa. Safe isolation of the petrous and paraclival portion of the internal carotid artery (ICA) is enabled as well [49,50,53].

Transoral robotic surgical (TORS) nasopharyngectomy is performed using a transpalatal wound, with the help of a high-resolution 3-D camera and two Endowrists. In the case of the involvement of the base of the skull, the transoral approach may be combined with the transnasal endoscopic approach in order to achieve a better treatment outcome [51]. Unfortunately, with the use of TORS nasopharyngectomy, the crucial tactile sensation is disabled, which may increase the risk of iatrogenic damage to the ICA [40,48,51].

With the use of ENP, negative resection margins were achieved in 73–96% of the cases, with the observed 2-year LC in 86%. Severe surgical complications occurred in 0–25% (osteonecrosis of the clivus, hypoxic brain damage, persistent palatal fistulae) and treatment-related deaths were observed in 0–6% [37,44,45,46,47,48,49,53] (Table 1). In a recent meta-analysis, 1-, 2-, and 5-year OS rates following salvage ENP were 97%, 92%, and 73%. A significant impact of T classification of the recurrent disease on prognosis was confirmed with the reported 2-year OS rate in T1, T2, T3, and T4 patients in 100%, 87%, 78%, and 38%, respectively [54].

During the COVID-19 pandemic, concerns regarding the safety of the endoscopic approaches to the nasopharynx arise as a high viral load of SARS-CoV-2 in the upper respiratory tract mucosa is documented [39]. The risk of aerosolization of blood and irrigation fluids is not negligible in this situation [55,56]. In order to minimize the risk, constant irrigation and smoke, fluid, and blood suction using a high-efficiency particulate air filter is recommended [57].

**Table 1 cancers-12-03510-t001:** Outcomes of endoscopic and maxillary swing nasopharyngectomies in the treatment of recurrent nasopharyngeal carcinoma (rNPC).

Study	Number of Patients	Surgical Approach	Median FU, Months	Treatment Outcome, %	Severe Complications, %	Treatment-Related Deaths, %
Chen [46] 2009	37	ENP	24	CM: 96 2-year OS: 84	0	0
Ko [44] 2009	28	ENP	13	CM: 89 2-year OS: 59	14	0
Sun [47] 2015	71	ENP	5-96	2-year OS: 74 5-year OS: 39	0	0
Tsang [48] 2015	12	ENP	24	CM: 75 2-year LC: 86 2-year OS: 83	25	0
You [37] 2015	72	ENP	49	5-year OS: 77	12	6
Liu [45] 2017	91	ENP	23	CM: 81 2-year OS: 65 5-year OS: 38	N/A	0
Chan [49] 2018	30	ENP	N/A	CM: 73	0	0
Wong [53] 2020	12	ENP	45	5-year OS: 50	0	0
Wei [58] 2011	246	MSNP	38	CM: 78 5-year LC: 74	21	0
Chan [59] 2012	312	MSNP	34	CM: 80 5-year LC: 74	16	0
Chan [60] 2015	338	MSNP	52	CM: 79 5-year LC: 74 5-year OS: 62	38	N/A

ENP, endoscopic nasopharyngectomy; MSNP, maxillary swing nasopharyngectomy; FU, follow-up; CM, clear microscopic margin; OS, overall survival; LC, local control; N/A, not available.

### 4.3. Maxillary Swing Nasopharyngectomy

Following maxillary swing nasopharyngectomy (MSNP), the observed 5-year LC and 5-year OS can reach up to 74% and 62%, respectively, with reported severe complications such as facial numbness, trismus, palatal fistulae, and otitis media with middle ear effusion in up to 38% of the patients [58,59,60] (Table 1). The maxillary swing approach involves a Weber Ferguson Longmire incision followed by osteotomies of the anterior and medial wall of the maxillary antrum and the lower portion of the zygomatic arch and separation of pterygoid plates from the maxillary tuberosity [40]. During the procedure, the maxillary osteocutaneous unit is unfolded and its blood supply is secured by the branches of the external carotid artery. With the use of a state-of-the-art MSNP, a complex exposure of the nasopharyngeal, oropharyngeal, and parapharyngeal space can be achieved and the acquirement of microscopically negative resection margins is potentially enhanced.

Throughout the MSNP, the ICA can be identified by manual palpation or with the use of intraoperative ultrasound. If there is ICA exposure, the surgeon may opt for a microvascular tissue transfer, which may result in a decreased risk of bleeding or carotid artery blow-out [40]. Tumors that encircle the ICA are technically challenging, but they are potentially resectable using an extracranial–intracranial bypass [61]. Therefore, the indication of MSNP can be extended to more clinical situations and MSNP may be safely offered to a carefully selected subset of patients with locally advanced T3-T4 rNPC [58,59,60,61,62].

### 4.4. Neck Dissection

Most regionally recurrent NPCs are technically resectable, and neck dissection (ND) is the treatment of choice for patients with isolated regional failure with the observed 5-year OS in 41% [63]. Currently, there are no uniform standards on the extent of the ND in this clinical setting.

Radical, modified radical, or selective neck dissection can all be considered for an isolated neck recurrence after initial RT. Radical neck dissection (RND) has long been considered the mainstay of treatment for the regional failure of NPC [64,65,66,67]. However, RND may worsen QoL of the patients because of the potential damage of the spinal accessory nerve and chronic shoulder stiffness and pain [68]. Whether such an extensive operation such as RND is truly necessary to achieve control of the neck disease is highly debatable, as cervical metastases in rNPC frequently present as a solitary node, and, sometimes, subsequent examination of the radical neck dissection specimen reveals no tumor in any of the lymph nodes removed [69]. Some authors suggest that as the neck nodes in level I are affected only in 4–5% of the patients with regionally recurrent NPC, a less extensive RND sparing level I might be considered [64,70,71].

Selective neck dissection (SND) may offer similar OS and disease-free survival (DFS) for patients with residual nodal disease in a single node following curative RT compared to patients that had undergone RND in this clinical setting, with the advantage of decreased peri- and postoperative morbidity [64,70]. Further studies are needed to determine and confirm which patients would benefit most from the SND procedure.

## 5. Re-Irradiation

Re-irradiation (overlap of the new radiation fields with the prior full-dose radiation volume) should be approached with caution in all cases, as there is a high risk of iatrogenic treatment-related morbidity and mortality [72]. Various techniques of re-irradiation have been proposed in this setting, including IMRT [17,19,20,37,73,74,75,76,77,78,79,80,81,82,83,84,85,86,87,88], SRS [89,90,91], stereotactic body radiotherapy (SBRT) [91,92,93,94,95,96,97,98,99,100], proton beam therapy (PBT) [101,102,103], carbon ion radiation therapy (CIRT) [104,105,106,107,108,109], and intracavitary or interstitial brachytherapy (BRT) [110,111,112,113,114,115,116]. It should be noted that any studies that support re-irradiation techniques in rNPC are small and retrospective in nature. They contain a limited number of patients with heterogenous stages and histological findings of the disease, and therefore data derived from these studies may be strongly biased. The therapeutic outcomes of the divergent RT approaches have never been compared in any prospective clinical trial. Hence, the treatment decision-making may depend upon the feasibility and local expertise, including the location and the extent of the recurrent disease and the estimated prognosis of the patient.

### 5.1. Recommendations for Radiotherapy during the COVID-19 Pandemic

To safeguard treatment capacity from staff shortages and mitigate the risk of patient infection by SARS-CoV-2 from daily hospital attendances, a large number of publications in the COVID-19 pandemic recommend consideration of hypofractionated radiation therapy (lower number of fractions with a higher dose per fraction) [117,118,119,120]. Indeed, although hypofractionated schemes have been proposed in head and neck cancer patients, the lowest currently accepted number of fractions for curative-intent treatments using IMRT according to the ASTRO-ESTRO Consensus Statement published in July 2020 is 20 [119]. However, if feasible and indicated, the choice to address the patient to mini-invasive endoscopic surgery would have the advantage to prevent patients from multiple accesses to the hospital, and the risk of cross-contamination would be limited [121].

### 5.2. IMRT

Intensity-modulated radiation therapy with its superior dose distribution and sparing of organs at risk (OaR) compared to the conventional 3-D conformal RT (3-D CRT) is the most commonly indicated modality of salvage re-irradiation in rNPC. However, in a recent meta-analysis, the reported grade 5 toxicity (treatment-related death) following re-irradiation based on conventionally fractionated IMRT in patients with rNPC was 33% [72]. In contrast to that, the potential for long-term survival with re-IMRT has been demonstrated by several authors in a carefully selected subset of patients [17,19,20,37,73,74,75,76,77,78,79,80,81,82,83,84,85,86,87,88].

The median prescribed dose of re-IMRT in the published studies is 60–70 Gy in 30–35 fractions to the GTV with an additional stereotactic radiosurgery (SRS) [75] or brachytherapy (BRT) boost [20]. In two series, hyperfractionation 1.2 Gy twice a day to the dose of 64.8 Gy was used [82,87]. Variable results of OS were observed (1-year OS 63–75%, 2-year OS 44–79%, 3-year OS 15–74%, 5-year OS 8–64%) [17,20,74,75,76,77,78,79,80,82,83,84,85,86] (Table 2). With the use of hyperfractionation, the median local failure-free survival showed a trend in favor of hyperfractionated re-IMRT compared to standard fractionation (28.2 vs. 16.6 months, *p* = 0.164) but OS was not significantly different (34.8 vs. 35.5 months, *p* = 0.603) [87]. 

With the use of re-IMRT, severe ≥ grade 3 toxicity was variably reported in 0–74%, predominantly in the form of hearing loss, cranial nerve palsy, temporal lobe necrosis, trismus, and dysphagia, in descending order. Treatment-related deaths were observed in 0–65% of the patients, which are alarming numbers that emphasize the need for very careful patient selection. However, these data should be interpreted with caution, as treatment mortality rates differ substantially between the Chinese studies [17,20,75,76,77,78,79,80,82,83] (Table 2) and studies from the rest of the world [74,84,85,86] (Table 2). In the series from endemic regions, the most common cause of treatment-related death was massive hemorrhage from the ulcerated nasopharyngeal mucosa, temporal necrosis, and cranial nerve palsy, in descending order. With the cautious patient selection and meticulous RT planning and delivery, which is a prerequisite for safe rNPC re-irradiation, a low rate of treatment-related mortality can be achieved [74,84,85,86] (Table 2).

Treatment-related hemorrhage was slightly reduced with the use of hyperfractionated re-IMRT compared to standard fractionation (30.0% vs. 0 %, *p* = 0.060) [87]. A model to predict the risk of lethal nasopharyngeal necrosis after re-IMRT in nasopharyngeal carcinoma patients was developed by Chinese authors [88]. It was concluded that female sex (*p* = 0.008), necrosis before re-irradiation (*p* = 0.008), accumulated total prescription dose to the GTV ≥145.5 Gy (*p* = 0.043), and recurrent tumor volume ≥25.38 cm^3^ (*p* = 0.009) were independent risk factors for lethal nasopharyngeal necrosis. Future scientific research may suggest strategies to prevent the development of severe complications following re-IMRT in patients struggling with rNPC.

**Table 2 cancers-12-03510-t002:** Outcomes of re-irradiation using intensity-modulated radiation therapy (IMRT) in the treatment of rNPC.

Study Country	Country	Number of Patients	Median FU, Months	Dose Range, Gy	Response, %	Late ≥ Grade 3 Toxicity, %	Grade 5 Toxicity, %
Chua [75] 2005	China	31	11	50–60	1-year OS: 63 CR: 58	25	N/A
Han [20] 2012	China	239	29	60–70	5-year OS: 45	70	35
Hua [76] 2012	China	151	40	60–70	3-year LC: 83 5-year LC: 81 3-year OS: 46 5-year OS: 38	34	0
Qiu [82] 2012	China	70	25	50–77	2-year OS: 67	36	N/A
Chen [77] 2013	China	54	16	50–77	1-year OS: 72 2-year OS: 44	48	25
Tian [80] 2014	China	251	40	60–70	5-year OS: 8–64	65	48
Tian [78] 2014	China	59 58	25	60 68	3-year OS: 38–57 5-year OS: 30–44	29 51	40 54
Chan [83] 2017	China	38	48	50–60	3-year LC: 44 3-year OS: 47	74	8
Tian [79] 2017	China	245	24	60–79	5-year OS: 27	27	29
Li [17] 2018	Singapore, China	558	36–41	60–70	3-year OS: 15–74	N/A	36–65
Roeder [84] 2011	Germany	14	20	36–64	2-year LC: 59 2-year OS: 44 3-year OS: 37	29	0
Puebla [85] 2015	Spain	17	23	50–70	2-year LC: 82 2-year OS: 79	0	0
Karam [86] 2016	Canada	27	36	40–60	2-year LC: 46 2-year OS: 49	37	0
Agas [74] 2019	Philippines	14	15	60	1-year OS: 75 2-year OS: 64	23	0

IMRT, intensity-modulated radiation therapy; Gy, gray; FU, follow-up; CR, complete response; OS, overall survival; LC, local control; N/A, not available.

### 5.3. Stereotactic Radiosurgery

Stereotactic radiosurgery is a highly accurate form of radiation therapy that delivers a single dose of precisely targeted radiation using focused gamma-ray beams that converge solely on the specific area where the tumor resides, minimizing the amount of radiation to healthy tissues. Limited evidence on the use of linac-based SRS [89] or Gamma Knife [90,91] as a single treatment modality in rNPC is available.

With the use of a single fraction of 12.5–18 Gy, satisfactory treatment outcomes were observed with reported 1-, 3-, and 5-year OS in 90–98%, 66–77%, and 47%, respectively [89,90,91] (Table 3). The time interval from previous RT, T classification of the recurrence, and tumor volume are all predictive factors of local control and survival. A prognostic scoring system for SRS in the treatment of rNPC was designed on the basis of these predictive factors. The 5-year local failure-free rate in patients with good, intermediate, and poor prognostic score was 100%, 42.5%, and 9.6%, respectively. The corresponding 5-year OS was 100%, 51.1%, and 0%, respectively [89].

Severe late toxicity reported in 0–33% of the patients after SRS for rNPC included brain necrosis, pituitary insufficiency, and cranial nerve palsy, in descending order, with no treatment-related deaths observed in the published studies [89,90,91].

**Table 3 cancers-12-03510-t003:** Outcomes of re-irradiation using non-IMRT techniques in the treatment of rNPC.

Study	Number of Patients	Technique	Median FU, Months	Median Dose, Gy	Response, %	Late ≥Grade 3 Toxicity, %	Grade 5 Toxicity, %
Chua [89] 2006	48	SRS	54	12.5	CR: 77 5-year OS: 47	27	0
Chua [91] 2009	43	SRS	40	12.5	1-year OS: 98 3-year OS: 66	33	0
Lee [90] 2020	10	SRS	18	18	1-year OS: 90 3-year OS: 77	0	0
Xiao [98] 2001	50	SBRT	20	24/6–8 fr	CR: 76 1-year OS: 84 2-year OS: 65 3-year OS: 60	18	16
Wu [99] 2007	90	SBRT	20	18/3 fr 48/6 fr	CR: 63–66	19	2
Chua [91] 2009	43	SBRT	24	34/2–6 fr	1-year OS: 78 3-year OS: 61	21	7
Leung [94] 2009	30	SBRT	47	54/18 fr	5-year OS: 40	57	3
Seo [95] 2009	35	SBRT	25	33/3–5 fr	CR: 72 5-year OS: 60	16	6
Ozyigit [96] 2011	24	SBRT	23	30/5 fr	2-year LC: 82	21	12
Dizman [97] 2014	24	SBRT	19	30/5 fr	1-year LC: 64 3-year LC: 21 1-year OS: 83 3-year OS: 31	4	4
Lin [102] 1999	16	PBT	24	59–70 (GyE)	2-year OS: 50 2-year LC: 50	31	0
Dionisi [103] 2019	17	PBT	10	60 (GyE)	1.5-year LC: 66 1.5-year OS: 54	23	0
Hu [108] 2018	75	CIRT	15	50–60 (GyE)	1-year OS: 98	12	1
Kwong [110] 2001	106	BRT ^198^Au grains	45	60	5-year LC: 63 5-year OS: 54	28	N/A
Shen [111] 2015	30	BRT ^125^I grains	38	130	CR: 81 2-year LC: 26 2-year OS: 30	N/A	N/A
Yan [112] 2017	39	BRT ^125^I grains	30	120	2-year LC: 41 2-year OS: 51	26	0
Leung [113] 2005	34	BRT HDR intracavitary	N/A	24 Gy/3 fr	5-year LC: 97 5-year OS: 78	29	N/A

SRS, stereotactic radiosurgery; SBRT, stereotactic body radiation therapy; PBT, proton beam therapy; CIRT, carbon ion radiation therapy; BRT, brachytherapy; HDR, high-dose rate; Gy, gray; GyE, gray-equivalent; fr, fractions; FU, follow-up; CR, complete response; OS, overall survival; LC, local control; N/A, not available.

### 5.4. Stereotactic Body Radiotherapy

Fractionated SBRT is a modification of SRS, which enables irradiation to be delivered in several fractions (typically 24–48 Gy/6 fr in rNPC) without losing the advantage of the mechanical precision and accuracy as well as dose conformity of SRS [92]. Published evidence suggests that tumor size reduction following ablative doses of RT could be largely dependent on T cell response [122]. Given that the activation of T lymphocytes in regional lymphatics is dramatically increased in response to ablative doses of RT, the treatment outcome could be enhanced by immunotherapy [93,122]. Furthermore, a common feature of EBV-associated NPC is the dense infiltration of lymphocytes in the tumor stroma and positive programmed death-ligand 1 expression in tumor cells, making it an especially attractive target for immune checkpoint inhibitors [123].Compared to SRS using a single fraction of high-dose irradiation, fractionated SBRT may be superior in terms of tumor control (3-year local failure-free rate 51% for SRS vs. 83% for SBRT, *p* = 0.003) but not OS (3-year OS in 66% for SRS and 61% for SBRT, *p* = 0.31). The incidence of severe late toxicity was 33% in the SRS group compared to 21% in the SBRT group [91]. The incidence of late ≥grade 3 toxicity in 4–57%, mostly in the form of nasopharyngeal, temporal lobe, and brainstem necrosis, was reported by other authors treating rNPC with SBRT [91,94,95,96,97,98,99] (Table 3), with treatment-related deaths occurring in 2–16%, mostly due to massive hemorrhage. Tumors involving the fossa of Rosenmüller and invading deeply to the foramen lacerum are the most important predisposing factors of fatal hemorrhage in this clinical setting [98].

Despite the significantly improved rate of severe toxicity and a decline in treatment-related deaths compared to re-IMRT in the treatment of rNPC, it cannot be definitely concluded that one technique is superior to the other; the available evidence on the use of SBRT is strongly biased by a short follow-up. The use of SBRT is generally recommended over re-IMRT for Ward’s RPA group III patients (<2 years from previous RT with organ dysfunction) [19]. This recommendation is based upon the observed 8 months median survival in this subgroup of patients with the aim to reduce overall treatment time in contrast to the 7-week IMRT. Albeit, no advantage in survival can be expected with the use of SBRT compared to re-IMRT [100].

### 5.5. Proton Beam Radiotherapy

The proton beam delivers the majority of its dose at the end of the range (Bragg peak) and almost no dose afterward, allowing a much better normal tissue sparing before and beyond the tumor region. High linear energy transfer (LET) protons are considered to be more effective than photons as they inflict more direct double-strands breaking DNA damage. A higher LET yields a higher relative biological effectiveness (RBE). Currently, most proton therapy treatments and studies assume a fixed RBE of 1.1 [101], but a biologically isoeffective dose is typically delivered to the patients, and therefore any definitive radiobiological advantage of protons is doubtful. The biological dose of proton/heavy-ion therapy is usually expressed as “Gy equivalent” (GyE) after taking the RBE into account.

Only a few authors report the use of PBT specifically in the treatment of rNPC [102,103] (Table 3). With the use of double scattering PBT, the 2-year OS and LC were both 50%. A significant association between adequate tumor coverage (defined as ≥90% of the treated volume receiving ≥90% of the prescribed dose) and OS was described [102]. With the use of active scanning PBT technique, the reported 1.5-year OS and LC were 54% and 66%, respectively. Severe late toxicity occurred in 23% of the patients in the form of hearing loss, dysphagia, and cervical vertebral osteoradionecrosis, in descending order [103]. More data and longer follow-up are required to refine the application of PBT in patients with rNPC.

### 5.6. Carbon Ion Radiotherapy

Carbon ion radiation therapy is still considered experimental for many tumor sites, but guidelines for its clinical indications are being established [104,105]. The accepted RBE for carbon ions used in clinical RT is generally estimated to be 2.5–3, however, values as high as 5 have been reported [106]. Complex DNA damage following CIRT, mostly in the form of double-strand breaks, is refractory to repair as it involves the clustering of multiple types of DNA lesions in close proximity to one another [109]. Given the physical and biological characteristics of carbon ions, CIRT has the potential to improve LC and reduce normal tissue complications in treating cancer patients, especially those with deep-seated and traditionally radioresistant tumors that have poor outcomes with standard photon therapies [107].

Initial clinical outcome on the use of CIRT in 75 patients with rNPC reported 1-year OS in 98% and 1-year local relapse-free survival in 87%. Severe toxicity was infrequent (12%) and included mucosal necrosis, xerostomia, and temporal lobe necrosis [108] (Table 3).

Carbon ion radiotherapy offers a promising alternative to patients with a short time interval from previous radical RT, and with its higher RBE and distinctive DNA damage, it bears the potential to cure even the most difficult-to-treat radioresistant tumors. However, more data and long-term follow-up is needed to refine the optimal dose and fractionation of CIRT in patients with rNPC.

### 5.7. Brachytherapy

The inherent physical and dosimetric characteristics of BRT allow the delivery of a very high dose to the nasopharynx while minimizing the dose to the adjacent structures. Current treatment options include brachytherapy alone with either interstitial [110,111,112] or intracavitary [113] approach, or in combination with external beam radiation therapy (EBRT) [114], SRS [115] and SBRT [116].

When permanent radioactive ^198^Au [110] or ^125^I [111,112] interstitial implantation was used, the reported 2- and 3-year LCs were 26–41% and 5–23%, respectively, and the reported 1-, 2-, 3-, and 5-year OS were 84%, 30–51%, 6–30%, and 54%, respectively [110,111,112] (Table 3). Moreover, LC was superior in recurrences confined to the nasopharynx compared to those extending beyond the nasopharynx [110]. Severe late toxicity occurred in 26%, mostly in the form of nasopharyngeal necrosis, epistaxis, and headache, with no treatment-related deaths observed [112]. Intracavitary high-dose-rate BRT predominantly used to treat T2 recurrences resulted in 5-year LC and OS in 97% and 78%, respectively, with severe late toxicity occurring in 29%, predominantly as endocrine dysfunction and cranial nerve palsy [113] (Table 3).

Outcomes of interstitial and intracavitary brachytherapy as single modalities in the treatment of small T1 and T2 recurrences may be comparable to ENP in terms of LC and OS and may be superior to re-IMRT concerning the reported rate of severe complications and treatment-related deaths [72]. However, published data on the use of BRT are very limited, and expertise in the procedure of radioactive seed implantation is not widespread and is generally restricted to highly specialized comprehensive cancer centers.

## 6. Conclusions

To the best of our knowledge, surgery or RT are the only curative treatment options for patients with recurrent non-metastatic NPC. The treatment of patients with rNPC is challenging, and precision radiation technologies such as SRS, SBRT, or BRT can be used effectively and safely to treat early stage and low-volume recurrent foci only. Conventionally fractionated IMRT is the most commonly indicated radiation treatment technique in this clinical setting. However, results from some series indicate that the use of re-IMRT is accompanied with a very high risk of radiation-induced morbidity and mortality. In the largest meta-analysis published to date, summarizing predominantly clinical experience from endemic areas, it was concluded that 33% of the patients with rNPC treated with re-IMRT died as a consequence of the treatment, whereas <30% died of local or distant failure (treatment period 2001–2012, 99% of analyzed cases coming from China) [72]. In this meta-analysis, however, due to issues with a moderate to high risk of bias in the methodological quality of the included studies, the overall quality of summarized evidence was deemed to be very low. Yet, a few series from the rest of the world provide reassurance of safe delivery of IMRT in terms of the risk of treatment-related mortality. Late morbidity of re-irradiation in the base of the skull region may be significant, and a high level of clinical and radiotherapy planning expertise is required. Despite limitations, given the widespread availability of the technology, IMRT seems to be the most feasible modality for the treatment of rNPC. In early local (T1, T2) and neck recurrences, retreatment with salvage surgery should always be considered in the view of low morbidity.

In the era of the COVID-19 pandemic, techniques of extreme hypofractionation (SRS, SBRT) and time-sparing interstitial or intracavitary BRT (if feasible) should be reconsidered as the risk of cross-contamination of both the patient and medical staff is definitely lower compared to 6–7 week normofractionated IMRT or heavy-ion therapy. We acknowledge that ENP yields superior results compared to IMRT in T1-T3 recurrent disease in terms of LC, QoL, and rate of severe complications. In the era of COVID-19, the advantages of surgical treatment are especially highlighted as it successfully prevents patients from multiple accesses to the hospital, and the risk of cross-contamination is probably the lowest with careful and repeated pre-operative SARS-CoV-2 virus testing.

Although the aim of this review was to present local treatment methods in the curative therapy of rNPC, the importance of comprehensive multidisciplinary treatment must be acknowledged. As new immunotherapeutic agents are being rapidly adopted in many cancers including NPC, promising response rates to the anti-programmed cell death-1 (PD1)-targeted therapy in the heavily pretreated NPC patients with locally recurrent disease have been demonstrated [124,125,126]. Future clinical trials evaluating a potential role in the curative treatment of rNPC are highly warranted [123]. Among the exciting scientific progress in the treatment of NPC, vascular endothelial growth factor inhibition and EBV-specific tumor antigens offer promising alternatives to patients with a locoregionally incurable disease.

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
