# Peer review of "Recurrent Nasopharyngeal Cancer: Critical Review of Local Treatment Options Including Recommendations during the COVID-19 Pandemic"

_cancers, 2020, doi:10.3390/cancers12123510_

Round 1

Reviewer 1 Report

Dear authors, this is an interesting and well-written review on recurrent NPC. only a couple of comments: I know that in this historic moment COVID-19 is an hot-topic but I don't see what it really add to your manuscript...

you should instead expand the section on systemic therapy and discussing immunotherapy for NPC (eg, https://www.redjournal.org/article/S0360-3016(19)32400-9/fulltext) 

https://academic.oup.com/jnci/article/111/7/655/5420162 etc)

Then, in MM section, you should improve the description of how literature search was performed, inclusion/exclusion criteria etc. While this is not a systematic review, many other types of manuscript exist and you must definetely clarify this methodological point

Reviewer 2 Report

The authors summarized lots of previous reports regarding the clinical options for  recurrent nasopharyngeal cancers. They digested the literatures for the readers to understand the treatment including salvage surgery and re-irradiation under COVID-19 pandemic situation. This review is well organized, and beneficial for head and neck oncologists. However, as the latest review article, the reference seems not to be enough comprehensive compared to the previous. Some points should be reconsidered.

Line 66-, early detection of the recurrence should include narrow band image (NBI) and should mention about usage of plasma EBV-DNA.

Line 192, severe complications are found in 0-40% of the patients. But Table 1 shows that severe complications of MSNP were 0% or N/A. 

In Table 1,2, and 3, the column of Response (LC or OS) are too busy to read. Letters or numbers should be separated a bit more from each other.

Table 1, Chan 2014 should be 2015.

Table 3, Leung 2015 should be 2005.

In references, many reports show no page ranges. For example, referene 1, CA Cancer J Clin. 2018; 68:394. Should be 394-424.

Line 471, 2011 should be 2012.

Line 643, The authors of Ref 77 are not correct. Y-M Tian, Y-H Tian, L Zeng, S Liu , Y Guan , T-X Lu, and F Han.

As for salvage surgery (Table 1), the following reports could be found through PubMed in addition to this article. What is the reason the authors excluded them?

  • King WW, Ku PK, Mok CO, Teo PM. Nasopharyngectomy in the treatment of recurrent nasopharyngeal carcinoma: a twelve-year experience. Head Neck 2000;22:215–22.

  • Hao SP, Tsang NM, Chang KP, Hsu YS, Chen CK, Fang KH. Nasopharyngectomy for recurrent nasopharyngeal carcinoma: a review of 53 patients and prognostic fac- tors. Acta Otolaryngol 2008;128:473–81.

  • Vlantis AC, Chan HS, Tong MC, Yu BK, Kam MK, van Hasselt CA. Surgical salvage nasopharyngectomy for recurrent nasopharyngeal carcinoma: a multivariate analysis of prognostic factors. Head Neck 2011;33:1126–31.

  • Bian X, Chen H, Liao L. A retrospective study of salvage surgery for recurrent nasopharyngeal carcinoma. Int J Clin Oncol 2012;17:212–7.

  • Ng LS, Lim CM, Loh KS. Long-term outcomes of nasopharyngectomy using partial maxillectomy approach. Laryngoscope 2016;126:1103–7.

  • Ko JY, Wang CP, Ting LL, Yang TL, Tan CT. Endoscopic nasopharyngectomy with potassium-titanyl-phosphate (KTP) laser for early locally recurrent nasopharyngeal carcinoma. Head Neck 2009;31:1309–15.

As for Table 2, the additional reports about re-irradiation using IMRT could be found through PubMed. They includes enough patients.

  • Hua YJ, Han F, Lu LX, Mai HQ, Guo X, Hong MH, et al. Long-term treatment outcome of recurrent nasopharyngeal carcinoma treated with salvage intensity modulated radiotherapy. Eur J Cancer 2012;48:3422–8.
  • Chen HY, Ma XM, Ye M, Hou YL, Xie HY, Bai YR. Effectiveness and toxicities of intensity-modulated radiotherapy for patients with locally recurrent nasopharyngeal carcinoma. PLoS One 2013;8:e73918.
  • Tian YM, Zhao C, Guo Y, Huang Y, Huang SM, Deng XW, et al. Effect of total dose and fraction size on survival of patients with locally recurrent nasopharyngeal carcinoma treated with intensity-modulated radiotherapy: a phase 2, single-center, randomized controlled trial. Cancer 2014;120:3502–9.
  • Tian YM, Huang WZ, Yuan X, Bai L, Zhao C, Han F. The challenge in treating locally recurrent T3–4 nasopharyngeal carcinoma: the survival benefit and severe late toxicities of re-irradiation with intensity-modulated radiotherapy. Oncotarget 2017.

As for Table 3, the additional reports about re-irradiation using stereotactic radiotherapy were found through PubMed. 

  • Dizman A, Coskun-Breuneval M, Altinisik-Inan G, Olcay GK, Cetindag MF, Guney Y. Reirradiation with robotic stereotactic body radiotherapy for recurrent nasopharyngeal carcinoma. Asian Pac J Cancer Prev 2014;15:3561–6.

  • Ozyigit G, Cengiz M, Yazici G, Yildiz F, Gurkaynak M, Zorlu F, et al. A retro- spective comparison of robotic stereotactic body radiotherapy and three-dimensional conformal radiotherapy for the reirradiation of locally recurrent nasopharyngeal carcinoma. Int J Radiat Oncol Biol Phys 2011;81(4):e263–8.

  • Seo Y, Yoo H, Yoo S, Cho C, Yang K, Kim MS, et al. Robotic system-based fractionated stereotactic radiotherapy in locally recurrent nasopharyngeal carcinoma. Radiother Oncol 2009;93:570–4.

  • Leung TW, Wong VY, Tung SY. Stereotactic radiotherapy for locally recurrent nasopharyngeal carcinoma. Int J Radiat Oncol Biol Phys 2009;75:734–41.

  • Chua DT, Wu SX, Lee V, Tsang J. Comparison of single versus fractionated dose of stereotactic radiotherapy for salvaging local failures of nasopharyngeal carcinoma: a matched-cohort analysis. Head Neck Oncol 2009;1:13.

Line 230, Ref 17, 34, 68, 78, 84, and 85 are not included in Table 2. 

Line 231, Ref 88, 89, 91, and 100 are not included in Table 3. 

Line 254, Ref 17,34, 78, 84, and 85 are not shown in Table 2. 

Line 261, Ref 17, 34, 68, 78, 84, and 85 are not included in Table 2.

This will be an issue of expression. These references are reviews or meta-analyses which should not be included in Table 2 or 3. But with these expression, readers can think they are discordance.

Line 259, What does BID stand for?

Line 370, What does EBRT stand for?

Line 380 and 407, What does ENPG stand for? ENP?

Line 392, The report of the largest meta-analysis should be added to references. Because the authors cited the results of the report.

Reference 100 can not be detected through PubMed. 

Round 2

Reviewer 2 Report

The authors addressed all reviewer's comments with high accuracy. The quality of their article has been improved precisely. I believe that this review article gives a lot of benefit to head and neck oncologists, indicating a load map to understand the treatments of recurrent NPCs.

Some minor points should be reconsidered.

In table 1 and 3, there are unnecessary bars under the first reports.

Line 268, IMRT [17, 19, 20, 37,, 78-93]. Commas are repeated.

Line 299, [, 17, 20, 79-85, 87-91, Table 2]. The first comma is not necessary.
